# Data-driven discovery of potent small molecule ice recrystallisation inhibitors

Matthew T. Warren [1,2,5], Caroline I. Biggs [1], Akalabya Bissoyi [3,4], Matthew I. Gibson [1,2,3,4] ✉ & Gabriele C. Sosso [1] ✉

Controlling the formation and growth of ice is essential to successfully cryopreserve cells, tissues and biologics. Current efforts to identify materials capable of modulating ice growth are guided by iterative changes and human intuition, with a major focus on proteins and polymers. With limited data, the discovery pipeline is constrained by a poor understanding of the mechanisms and the underlying structure-activity relationships. In this work, this barrier is overcome by constructing machine learning models capable of predicting the ice recrystallisation inhibition activity of small molecules. We generate a new dataset via experimental measurements of ice growth, then harness predictive models combining state-of-the-art descriptors with domain-specific features derived from molecular simulations. The models accurately identify potent small molecule ice recrystallisation inhibitors within a commercial compound library. Identified hits can also mitigate cellular damage during transient warming events in cryopreserved red blood cells, demonstrating how data-driven approaches can be used to discover innovative cryoprotectants and enable next-generation cryopreservation solutions for the cold chain.

The low-temperature storage of cells, tissues and biologics is an essential technology for biomedical research, regenerative medicine and vaccination distribution[1,2]. Biologics are most effectively preserved at sub-zero temperatures, however there are several stressors associated with these conditions, originating from the formation (nucleation) and growth (recrystallisation) of ice[3]. In nature, many organisms can mitigate the harmful effects of ice by accumulating solutes, which provide a cryoprotective effect on a colligative basis[4]. Cold-adapted species may also produce antifreeze proteins and macromolecules that can recognise and bind to nascent ice crystals directly, inhibiting growth at the bound surface[5]. This process, known as ice recrystallisation inhibition (IRI), has drawn significant interest due to its potential to enhance the cryopreservation of cells and tissues ex vivo. To this end, a variety of biomimetic materials have been explored, including synthetic protein analogues[6], polymers[7] and self-assembling compounds[8]. More recently, small molecules have also been found to slow the recrystallisation of ice via an alternative mechanism that does not involve binding directly to the crystal surface, improving the cryopreservation outcomes for human blood and stem cells, as well as mammalian organs[9–11]. Nonetheless, cryopreservation protocols still employ significant quantities of cosolvent (e.g. DMSO) which can have several adverse effects, including toxicity and epigenetic alterations[12,13]. Moreover, the diversity of cell characteristics and storage requirements mean that rarely does any single cryoprotectant provide optimal outcomes across different settings[3,14]. Consequently, there is a pressing need for novel ice recrystallisation inhibitors (IRIs) for a number of applications, most notably cell-based therapies which require cryopreservation throughout their supply chains[2].

Despite significant efforts, identifying new IRIs remains a formidable task. Guided by limited mechanistic understanding, as well as structure-activity relationships that can be counterintuitive and contradictory, the discovery of new materials is fundamentally trial and error. Traditionally, the synthesis and screening of hundreds of compounds has been required in order to identify a handful of potent

[1]Department of Chemistry, University of Warwick, Coventry, UK. [2]Warwick Medical School, University of Warwick, Coventry, UK. [3]Manchester Institute of Biotechnology, University of Manchester, Manchester, UK. [4]Department of Chemistry, University of Manchester, Manchester, UK. [5]Present address: Institute of Cancer Research, London, UK. ✉e-mail: matt.gibson@manchester.ac.uk; g.sosso@warwick.ac.uk

'hits'[15]. This process is time-intensive, compounded by a lack of high-throughput techniques to quantitatively assess IRI, and has thus become the bottleneck in discovering IRI-active molecules to serve as next-generation cryoprotectants. In our previous work, we introduced amino acids as a novel class of IRIs that can be highly effective at low (millimolar) concentrations[16,17]. Benefiting from their low-cost, chemical diversity and commercial availability, amino acids are poised for industrial application as cryoprotectants. In this article, we introduce a machine learning (ML)-based approach for accurately predicting the IRI activity of amino acids and their derivatives. We first generated a dataset of these small molecules via large-scale experiment assessment of IRI, and compiled additional datasets from IRI activity measurements reported in literature. Trained using this data, we evaluated the performance of a variety of ML models featuring state-of-the-art descriptors, as well as novel representations computed via atomistic molecular dynamics (MD) simulations targeting interactions between hydrated inhibitors and water. To overcome the challenges of limited training data, we leveraged this descriptive information via an ensemble of models with different inputs and architectures, leading to more robust predictions with uncertainty estimates. We then applied this model to screen a commercial compound library and validated its predictions experimentally, revealing non-obvious small molecule IRIs that improved post-thaw cell viability of cryopreserved red blood cells (RBCs) exposed to transient warming events. Overall, this data-driven approach signals a fundamental shift from trial-and-error towards the rapid and efficient discovery of IRI-active materials for applications in industrial and clinical sciences.

## Results

### Classifying IRI-active small molecules

To benchmark our methods against earlier work, we first developed classification models to predict IRI activity on a categorical basis. Previously, Briard and coworkers[15] constructed a binary classification model to predict whether small-molecule carbohydrates were IRI-active or IRI-inactive using the training data denoted here as the Glyco dataset (Table 1, Fig. 1a). IRI activity was measured using the 'splat cooling' assay (SCA), wherein the average ice crystal (grain) size is determined following a period of recrystallisation (Fig. 1d, see Methods). This assay yields data in the form of mean grain size (MGS) values, which are typically normalised against a negative control for inhibition, producing a relative (%) MGS metric. Hence, for classification, percentage MGS (% MGS) values must be converted to 'active'/'inactive' class labels according to a given threshold, chosen to be 70 % MGS by Briard and colleagues[15].

To construct classification models, we employed six different chemical representations including low- and high-dimensional molecular descriptors, as outlined in the Methods. Neural network (NN) classifiers were then trained and validated using the Glyco dataset. Details of model architectures and training procedures are described

in Methods and Supplementary Section 2.2.4. Model performance was evaluated using the same metrics as ref. 15, also defined in the Supplementary Section 2.2.1. The performance of the classification models in terms of these metrics are displayed in Table 2.

The performance of these models was in line with the previous benchmark, with most achieving an F-score of 0.64 or higher, defined as the harmonic mean of precision and recall. Given that there were a number of individual models, each showing satisfactory performance, we sought to combine their knowledge by aggregating the predictions as illustrated in Fig. 1e. This approach, known as ensemble learning, is fundamental to certain supervised learning algorithms (e.g. random forests[18]) and has also proved successful for chemical property prediction in other domains[19]. The results of the top three performing ensembles, shown in Table 2, revealed improvements across most metrics compared to any individual model. Optimal results were obtained when all (six) models were used (Ensemble 2, Table 2), despite the fact that contributing models had weak performance individually. Although the specificity (0.68) of these ensemble predictions was in certain cases lower than other individual models, the high precision (0.82) represents an improvement compared to the previous benchmark (Table 2). We also emphasise that high precision can be advantageous in applications such as this, wherein the primary objective is to accelerate the discovery of (IRI-)active materials while avoiding false positives.

However, despite these encouraging results, we argue that predicting whether a given molecule belongs to the 'active' or 'inactive' class is not the optimal framework for this task. As there is no clear definition for an 'active' small molecule IRI, the modeller's choice of threshold is arbitrary, but in fact can have a significant impact on the prediction accuracy. In Fig. 1b, c, we highlight the effects of this threshold on the numbers of 'active' and 'inactive' molecules, and the performance of the corresponding ensemble model. Across these thresholds, the best scores were obtained using a value of 70 % MGS—the threshold used in the previous benchmark[15]—resulting in a slight positive imbalance of active labels (Fig. 1b). Although this model had good predictive power, it has limited practical utility, providing no insight into the degree of IRI-activity, bearing in mind that only the most active compounds are of interest with respect to a cryoprotective application. Instead, we suggest that this task is better formulated as a regression problem, wherein the model is trained to predict absolute % MGS values.

### Predicting absolute IRI activity

Stepping away from the classification framework described in the previous section, we trained and evaluated NN regressors using each descriptor independently (see Methods and Supplementary Section 2.2.4). At this point, we switched to using the Glyco2 dataset, which includes the entire Glyco set, as well as 99 additional carbohydrates tested for IRI activity under the same conditions (Table 1). These

## Table 1 | Datasets featured in this work

| Dataset | Purpose | N | Compound classes | Source |
|---|---|---|---|---|
| Glyco | Train/validate | 124 | Aryl glycosides, aryl/alkyl aldonamides | Ref. 15 |
| Glyco2[a] | | 223 | Aryl/allyl/amino glycosides, aryl/alkyl aldonamides, mono/disaccharides, lysine-based surfactants, cationic anti-INAs | Refs. 10,15,20,30–35 |
| Amino | | 63 | (α/β/γ-)amino acids/alcohols/esters | This work |
| Combined[b] | | 286 | See above | See above |
| Amino library | Screen | 497 | (α/β/γ-)amino acids/alcohols/esters | This work |
| Amino prediction | Test | 17 | (α/β/γ-)amino acids/alcohols/esters | This work |

Complete datasets including compound names, SMILES and % mean grain size (MGS) values are available at https://github.com/gcsosso/DOLMEN.
[a]Includes compounds in Glyco.
[b]Includes compounds in Glyco2 and Amino.

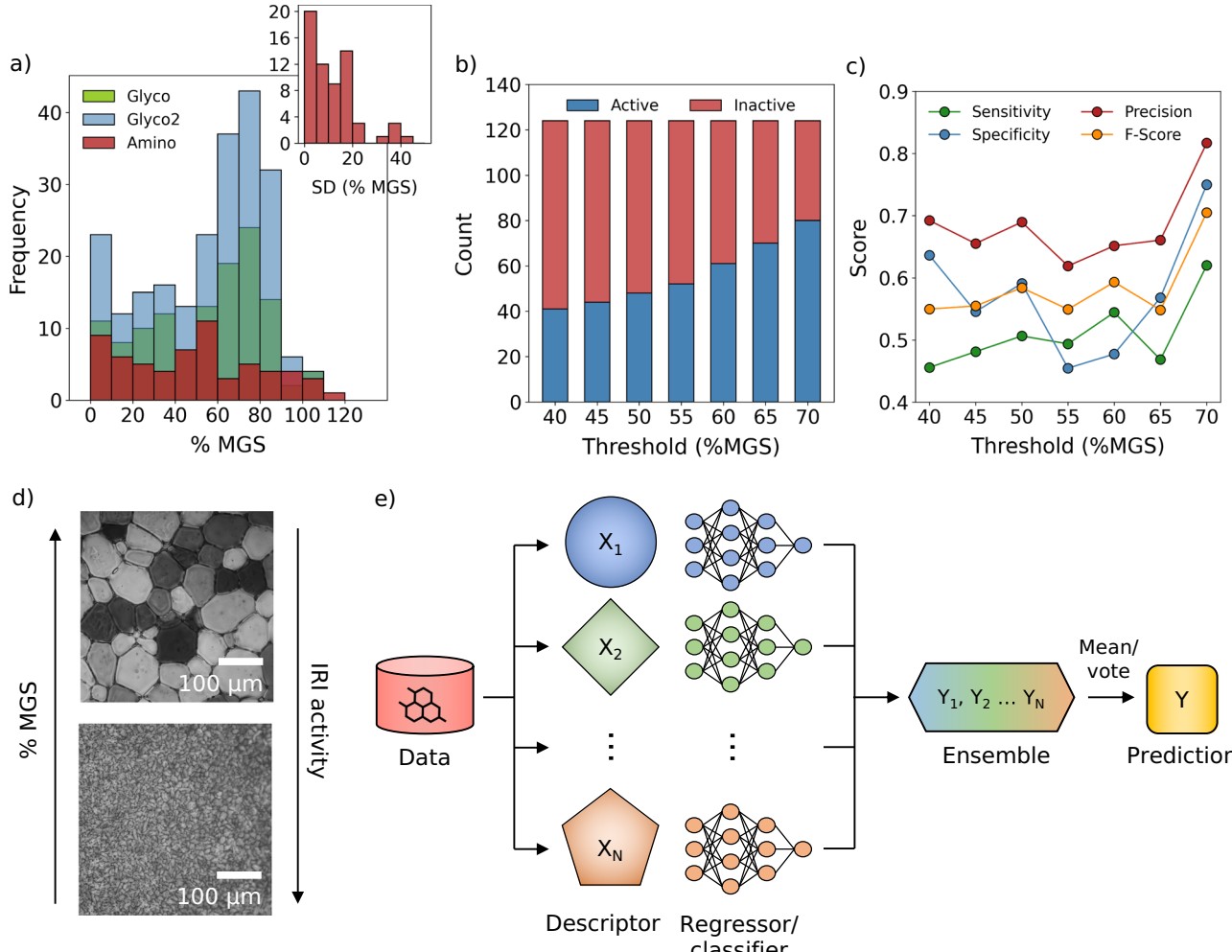

**Fig. 1 | Datasets and data-driven approaches. a** Frequency histograms showing the distribution of % mean grain size (MGS) values across the different datasets. The inset shows the distribution of the error, reported as the standard deviation (SD), for the Amino dataset. **b** The effects of the choice of the activity threshold on the proportion of compounds labelled 'active' or 'inactive' in the Glyco dataset. **c** Performance of the ensemble classification model using all six representations, trained and tested on the Glyco dataset. **d** Cryomicrographs obtained from the 'splat cooling' assay performed on exemplary ice recrystallisation inhibition (IRI)-inactive (top) and IRI-active (bottom) materials, whereby a smaller % MGS value corresponds to a more active inhibitor. **e** Schematic illustrating the ensemble learning approach. Different representations or descriptors are computed and used to train independent models, whose outputs are combined via majority voting scheme (classification) or mean average (regression). Source data are provided as a Source Data file.

structures are sufficiently similar to enable efficient training, while also increasing the chemical diversity of the dataset (Supplementary Fig. S4). In addition, we have also assembled a new dataset—Amino—via our own experimental measurements of 63 amino acids, shown previously to be a novel class of IRI-active material[16,17] (Table 1). IRI activity was evaluated using the SCA, however different solution conditions were used which can affect IRI activity. This is discussed in greater detail in the Methods and Supplementary Section 1.1.

Separate regressors were trained and evaluated using the Amino and Glyco2 datasets independently and in combination (Combined dataset). The results of these models, reported using mean squared error (MSE) and Pearson correlation coefficient (PCC) metrics, are shown in Table 3. Individually, most models had a relatively high predictive error, with experimental and predicted % MGS values showing only moderate correlation. Exceptions included the smooth overlap of atomic positions (SOAPs) and hydration indices models, whose predictions were well correlated with the experimental values for the Amino dataset (PCC ≥ 0.60). Overall, these results were expected given the high degree of uncertainty associated with the SCA, where significant variation can be observed between measurements, as shown by the error distribution in the inset of Fig. 1a. This uncertainty limits the maximum

expected correlation and minimum expected error of the model to ≈0.90 and ≈200, respectively (Supplementary Fig. S3 and Supplementary Section 2.2.5). A further discussion of SCA measurement error and confidence intervals is provided in Supplementary Section 1.3.

Given the success of the classification ensembles, we again explored an ensemble approach by obtaining the mean prediction across different combinations of three or more individual models. For regression, ensembles also provide a means to quantify the uncertainty associated with each prediction (e.g. as the standard deviation (SD) of the predictions). The results for the top three performing ensembles, reported in Table 3, highlight improvements in prediction accuracy with respect to each dataset. This was particularly significant in the case of the Amino dataset, with the ensemble test predictions achieving a correlation of 0.72. Whereas the best performing classification ensembles combined the predictions from five to six individual models, the optimal regression ensembles comprised three or four models. This reflects the fact that a numerical average is more susceptible to the effects of anomalous predictions compared to a majority-voting consensus. Similarly, computing the mean of many predictions can have the effect of 'smoothing out' the predicted values, especially at the tails of the target distribution.

**Table 2 | Performance metrics for test predictions obtained from classification models trained and evaluated on the Glyco dataset, using an activity threshold of 70% mean grain size (MGS)**

| Model | Sensitivity | Specificity | Precision | F-score |
|---|---|---|---|---|
| Benchmark[a] | 0.67 | 0.80 | 0.67 | 0.67 |
| Standard descriptors | 0.59 | 0.66 | 0.76 | 0.67 |
| Molecular cliques | 0.78 | 0.46 | 0.72 | 0.74 |
| H-wACSFs | 0.62 | 0.43 | 0.66 | 0.64 |
| SOAPs | 0.65 | 0.55 | 0.72 | 0.68 |
| Hydration histograms | 0.54 | 0.71 | 0.77 | 0.64 |
| Hydration indices | 0.38 | 0.77 | 0.75 | 0.50 |
| Ensemble 1[b] | 0.68 | 0.68 | 0.79 | 0.74 |
| Ensemble 2[c] | 0.62 | 0.75 | 0.82 | 0.71 |
| Ensemble 3[d] | 0.67 | 0.61 | 0.76 | 0.71 |

Metrics were computed for test predictions obtained via a single leave-one-out cross validation run, however, we show that repeating this process generates very similar scores (Supplementary Table 7), hence we report only the values for a single representative model.
[a]Data obtained from ref. 15.
[b]Ensemble using all representations except hydration histograms.
[c]Ensemble using all representations.
[d]Ensemble using all representations except hydration histograms and indices.

Figure 2 shows the predictions of the best ensemble models versus the experimental % MGS values for the Amino, Glyco2 and Combined datasets. These data, along with results shown in Table 3, highlight that models trained and evaluated using the Amino dataset yielded predictions with significantly higher correlation and lower error with respect to measured values, compared to the Glyco2 or Combined datasets. This was surprising given that the Glyco2 dataset was significantly larger than Amino, providing more examples to learn from during training. For the Combined dataset, the poor performance may be a product of the chemical disparity between the amino acids and carbohydrates. Indeed, Tanimoto similarities computed for these two distinct sets of compounds revealed significant structural diversity (Supplementary Fig. S5), while dimensionality reduction performed on the molecular descriptors used in this work showed that these classes of molecules span different regions in feature space (Supplementary Fig. S6). However, it remains unclear why Glyco2 models could not achieve scores on par with the Amino: possible explanations include experimental inconsistencies within this literature dataset, or a failure of the representations to capture the important structural features for this class of molecule.

## Hydration numbers and indices

Individually, the hydration indices models gave the most striking results, achieving good performance for the Amino dataset (PCC = 0.63), but performing poorly when used in conjunction with the Glyco2 or Combined datasets. The concept of a hydration index was first introduced by Tam and coworkers to explain the hydration-dependent IRI activity observed in carbohydrates, whereby more hydrated molecules showed greater levels of inhibition[20]. In the original formulation, hydration numbers were derived from molar compressibility coefficients according to the Passynsky equation[21], while the molar volume was calculated from density measurements[22]. In this work, hydration numbers were determined by finding the average number of water molecules within a given cutoff distance from the inhibitor molecule determined via MD simulations, as described in the Methods.

This quantity provides an estimate of the number of waters associated with each inhibitor, with a cutoff distance that can be tuned

to capture information about the system over different ranges. To account for the molecule's size, we then normalised this value against different quantities (e.g. volume or solvent-accessible surface area, SASA) also computed from MD simulations (see Methods), instead of molar volume which was used in the original work[20]. Hydration numbers were calculated using a number of fixed cutoffs as well the distance corresponding to the minima of the first and second solvation shells, determined from the hydration histograms illustrated in Supplementary Fig. S14. The latter cutoffs are tailored to each molecule individually, giving hydration numbers which approximate the number of water molecules in the first and second solvation shells surrounding the solute, respectively. We also included the number of hydrogen-bonded waters as an alternative for the hydration number.

As expected, using a greater cutoff distance produced a greater hydration number and index for a given compound, as well as a broader distribution of these values for each dataset collectively (Supplementary Fig. S7). We also found that the different hydration numbers and indices are correlated to varying degrees with IRI activity (% MGS), and that these trends were not shared between the Amino and Glyco2 datasets. This data is summarised in Fig. 3. We observed negative correlations between the hydration numbers and % MGS values for both datasets when a distance cutoff greater than 0.3 nm was used, which includes the distance corresponding to the second solvation shell. Yet, when these numbers were normalised against different size metrics to yield hydration indices, we found positive correlations with the Amino dataset and few correlations for Glyco2 (Fig. 3). For the Amino dataset, a positive correlation was observed for nearly all the different hydration indices, producing the strongest correlation when normalised by the molecular volume or weight. Meanwhile, only the indices calculated using hydrogen bonding data showed any correlation for the Glyco2 dataset, following the same positive trend. The correlations between IRI activity and size metrics individually are provided in the Supplementary Information.

Tam and colleagues previously showed that experimentally-derived hydration indices for a set of nine mono- and disaccharides gave a strong negative correlation with MGS measurements[20]. This finding gave support to a hypothesised IRI mechanism, wherein compounds with greater hydration (indices) cause more disruption to the ordering of surrounding quasi-liquid layer (QLL)/bulk water, increasing the energy associated with the transfer of bulk water to a growing ice crystal via the QLL and consequently slowing ice growth[20]. Our computational hydration indices, both for this same set of nine sugars, as well as the entire Glyco2 and Amino datasets, did not reproduce this correlation, irrespective of the cutoff distance used: instead, we observed a moderate positive correlation with % MGS values (PCC ≈ 0.5) for the Amino dataset, and limited correlation for the Glyco dataset (PCC ≈ 0.3). Note that a positive correlation indicates that more 'hydrated' molecules display weaker IRI.

Although our results are not in agreement with previous findings, these correlations do explain why the hydration indices were an effective ML descriptor for the amino acids, but not the carbohydrates. Moreover, is important to emphasise that whilst our computational hydration index is inspired by the previous work of Tam and colleagues[20], the values obtained using our method are not equivalent to those calculated using experimental data. Indeed, when the computational and experimental indices were compared for the set of mono- and disaccharides examined by Tam et al. [20], they showed only weak correlation (Supplementary Fig. S8). However, in light of limited experimental molar volume and compressibility coefficient data, a computational approach allows a hydration index to be calculated for virtually any chemical structure. Our analysis therefore included ~300 data points, encompassing highly active to inactive materials, compared to nine relatively inactive carbohydrates investigated in the aforementioned work. The size of these samples makes it challenging to draw comparative conclusions, as it is possible the previously

**Table 3 | Performance metrics for test predictions obtained from individual and ensemble regression models**

| Dataset | Model | Std. descriptors | Molecular cliques | H-wACSFs | SOAPs | Hyd. histograms | Hyd. indices | MSE | PCC |
|---|---|---|---|---|---|---|---|---|---|
| Glyco2 dataset | Standard descriptors | ✓ | | | | | | 631 | 0.39 |
| | Molecular cliques | | ✓ | | | | | 699 | 0.39 |
| | H-wACSFs | | | ✓ | | | | 579 | 0.44 |
| | SOAPs | | | | ✓ | | | 719 | 0.25 |
| | Hydration histograms | | | | | ✓ | | 682 | 0.33 |
| | Hydration indices[a] | | | | | | ✓ | 666 | 0.24 |
| | Ensemble G1 | ✓ | ✓ | ✓ | | | | 576 | 0.46 |
| | Ensemble G2 | ✓ | ✓ | ✓ | ✓ | | | 575 | 0.46 |
| | Ensemble G3 | | ✓ | ✓ | ✓ | | | 576 | 0.46 |
| Amino dataset | Standard descriptors | ✓ | | | | | | 673 | 0.59 |
| | Molecular cliques | | ✓ | | | | | 966 | 0.45 |
| | H-wACSFs | | | ✓ | | | | 738 | 0.54 |
| | SOAPs | | | | ✓ | | | 624 | 0.64 |
| | Hydration histograms | | | | | ✓ | | 960 | 0.37 |
| | Hydration indices[a] | | | | | | ✓ | 667 | 0.63 |
| | Ensemble A1 | ✓ | | | ✓ | | ✓ | 503 | 0.72 |
| | Ensemble A2 | | | ✓ | ✓ | | ✓ | 509 | 0.71 |
| | Ensemble A3 | | ✓ | ✓ | ✓ | | ✓ | 521 | 0.71 |
| Combined dataset | Standard descriptors | ✓ | | | | | | 642 | 0.43 |
| | Molecular cliques | | ✓ | | | | | 691 | 0.45 |
| | H-wACSFs | | | ✓ | | | | 644 | 0.45 |
| | SOAPs | | | | ✓ | | | 687 | 0.42 |
| | Hydration histograms | | | | | ✓ | | 760 | 0.37 |
| | Hydration indices[a] | | | | | | ✓ | 775 | 0.22 |
| | Ensemble C1 | ✓ | ✓ | ✓ | | | | 561 | 0.53 |
| | Ensemble C2 | ✓ | ✓ | ✓ | | ✓ | | 570 | 0.52 |
| | Ensemble C3 | ✓ | ✓ | ✓ | | | | 581 | 0.52 |

For the ensemble models, tick marks indicate the models included in the ensemble. As with the classification models, we report the metrics computed on a complete set of test predictions generated via leave-one-out cross validation. We note that very similar results are produced if this process is repeated (Supplementary Table 6).
*H-wACSFs* histograms of weighted atom-centred symmetry functions, *SOAP* smooth overlap of atomic position, *MSE* mean squared error, *PCC* pearson correlation coefficient.
[a]Multiple hydration indices with different hydration numbers used.

reported trend may not hold for a larger set of small molecules, especially more active inhibitors. Overall, our computational hydration numbers and indices, derived in a different fashion but using similar intuition to the experimental properties, suggest a different interpretation. While it is not possible to draw a causal relationship between hydration parameters and the mechanism of IRI, these results challenge the notion that hydration is a robust correlate of IRI for small molecules, and call for a review of this property and the hydration hypothesis.

**Discovering amino acid IRIs**
Returning to the predictive ML models, optimal performance was achieved using an ensemble that combined the predictions from three models trained using standard descriptors, SOAPs and hydration indices, in conjunction with using the Amino dataset (Ensemble A1, Table 2, MSE = 502, PCC = 0.72). Encouraged by these results, we sought to use our model to computationally screen a set of untested amino acids for IRI activity. To achieve this, we compiled a chemical library consisting of ≈500 amino acids (Table 1), as outlined in the Methods. These structures were all commercially available, hence no chemical synthesis was required, which is a significant and deliberate

advantage of this discovery platform. % MGS predictions for each compound in this library were obtained using an ensemble model trained on the entire Amino dataset. Given that the hydration indices require a MD trajectory for their calculation, we opted to use an ensemble comprising the standard descriptors, molecular cliques and SOAPs. Compared to the best model (Ensemble 1, Table 2), this model yielded predictions with a similar correlation (PCC = 0.67) and error (MSE = 570) when evaluated on the test set (Table 3), but has the advantage that these representations can be computed directly from a molecule's SMILES code at minimal computational cost (see Methods). These predictions were then ranked, and the ten most and ten least active molecules were designated as the prediction set (Table 1), taking into account other practical considerations such as their cost and availability. The distributions of predicted % MGS values and associated error (SD) for the entire prediction library are shown in Supplementary Fig. S9. Compounds with both low and high % MGS predictions were included in the final prediction set to assess the model's accuracy in predicting IRI across a range of activities, despite the primary objective being to identify highly active inhibitors.

To verify these predictions experimentally, the prediction set was then tested at an initial concentration of 20 mM in 10 mM NaCl using

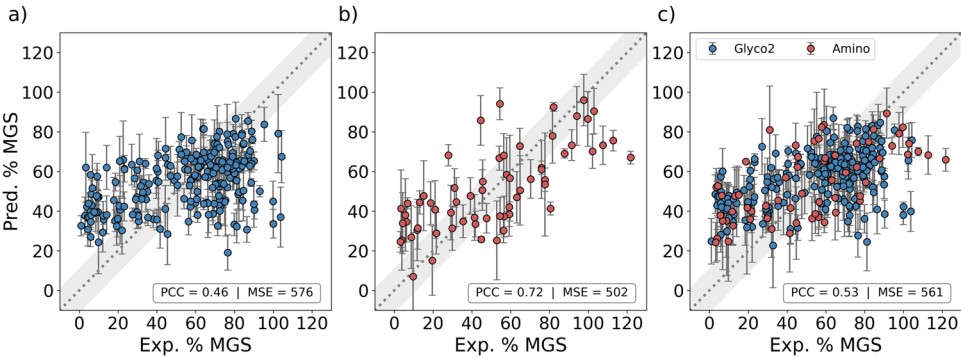

**Fig. 2 | Performance of cross validated ensemble models.** Scatter plots showing the % mean grain size (MGS) predictions from the best performing ensembles (G1, A1 and C1) trained and evaluated using: (**a**) the Glyco2 dataset; (**b**) the Amino dataset; and (**c**) the Combined dataset. The models comprising these ensembles are shown in Table 2. Data are presented as mean values ± the standard deviation (SD) of the individual test predictions from leave-one-out cross validation (predicted) or three independent repeats (experimental). The insets report the Pearson correlation coefficient (PCC) and the mean squared error (MSE). Grey shaded region represents the typical error associated with experimental % MGS measurements, taken as the mean SD of three measurements for all compounds in the Amino dataset. Source data are provided as a Source Data file.

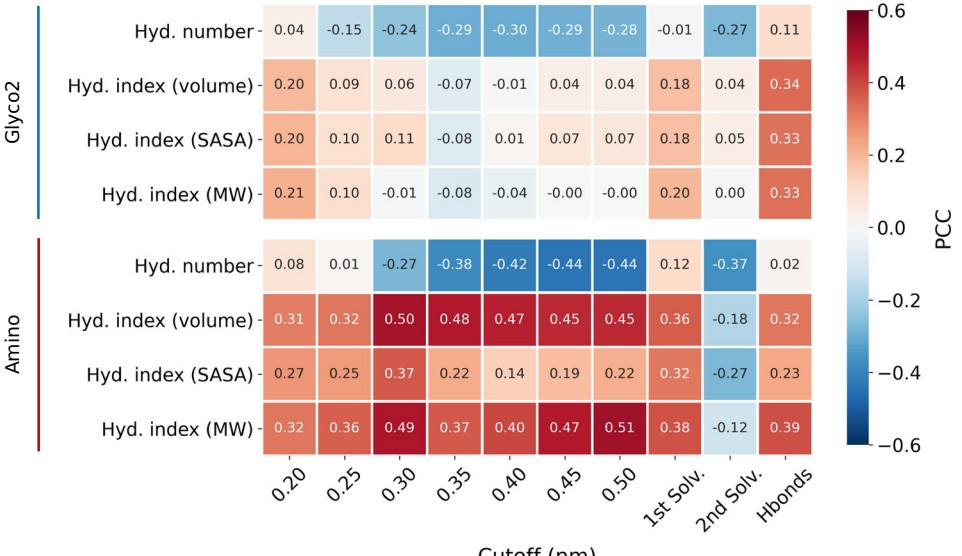

**Fig. 3 | Hydration property heatmap.** Heatmap showing the Pearson correlation coefficient (PCC) between % mean grain size (MGS) values and hydration numbers or hydration indices calculated using different cutoffs and size metrics for the Amino and Glyco2 datasets. Hydration indices were obtained by dividing the hydration numbers by either the molecular volume, the solvent accessible surface area (SASA) or the molecular weight (MW), as indicated. Only the hydration numbers normalised against molecular volume were used to train classification and regression models. Source data are provided as a Source Data file.

the SCA. Three compounds (compounds **7**, **8** and **9**) were tested at amino acid concentration of 10 mM, as they were insoluble at the initial concentration. A further three compounds could not be tested due to poor solubility. The predicted and experimental % MGS values for the prediction set are shown in Fig. 4a. Overall, the model achieved excellent performance, with 13 out of 17 predictions being in close agreement with the true % MGS, taking into account the uncertainty associated with the ensemble predictions and experimental measurements. This corresponded to a PCC of 0.61 and MSE of 483, similar to the level of performance observed for the test set during cross validation (Table 3).

For the molecules that were predicted to be highly active (% MGS < 20), only three (**1**, **3** and **5**) out of nine had moderate IRI activity (% MGS ≈ 50) when tested experimentally at 20 mM (Fig. 4a). These compounds all featured an aminopyridine scaffold with an O-methylated carboxylic acid (methyl ester) group adjacent to the pyridine nitrogen (Fig. 4b and Supplementary Fig. S10). Interestingly, these same moieties are present in compound **4**, which was significantly more active than **1**, **3** or **5**. Other compounds correctly predicted to be highly active were chlorinated L-tyrosine derivatives **6** and **8**, nitropyridine carboxylic acid **9** and aminooxazole esters **2** and **7** (Fig. 4b and Supplementary Fig. S10). The most active hit among this set was the aminooxazole ester **7**, which was able to prevent growth almost entirely at 10 mM. Dose-dependency experiments revealed that this compound retained IRI activity at 2.5 mM (0.4 mg/mL) when tested in NaCl, and was also active at 10 mM in phosphate-buffered saline (PBS) (Supplementary Figs. S15 and S16). PBS is an isotonic solution routinely used in biological applications, hence maintaining IRI activity under these conditions is typically required for cryopreservation. Ice shaping assays performed using a nanolitre osmometer also revealed crystal morphologies with no branching or faceting, confirming that this compound does not alter the growth habit of individual ice crystals (Supplementary Fig. S20)[23]. This suggests no direct binding to the ice crystal surface, as observed for other small molecules[16,20].

Compared to the training data, we note that compounds **2**, **4**, **7** and **13** represent some of the most IRI-active amino acids discovered to date; less than 10 % of all amino acids surveyed have an IRI activity below 10% MGS (Supplementary Fig. S19). It is important to clarify that

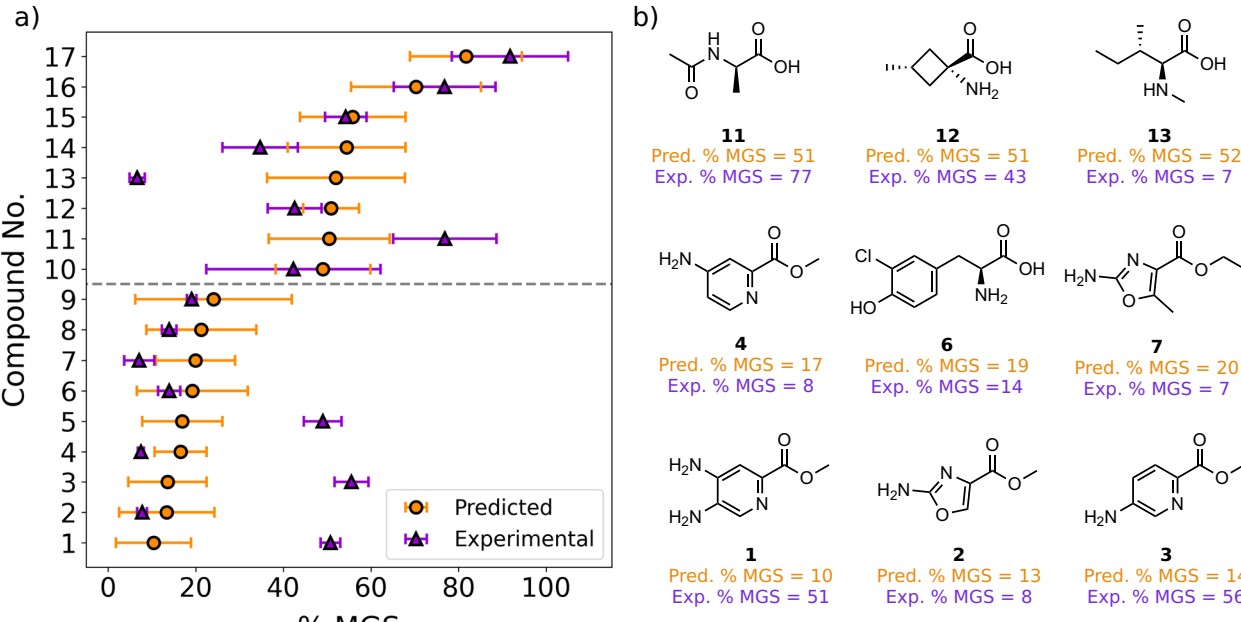

**Fig. 4 | Ensemble predictions for novel ice recrystallisation inhibitors.**
**a** Comparison of the predicted and experimental % mean grain size (MGS) values for the 17 compounds in the prediction set. Compounds were tested at 20 mM in 10 mM NaCl, with the exception of compounds **7**–**9**, which were tested at 10 mM.

Data are presented as mean values ± the standard deviation (SD) of the individual model predictions (predicted) or three independent repeats (experimental). **b** Chemical structures of a selection of compounds in the prediction set. Source data are provided as a Source Data file.

the IRI-active materials identified here do not represent entirely novel chemical scaffolds, bearing some resemblance to molecules in the training set, which can be quantified by assessing their Tanimoto similarity (Supplementary Fig. S11). We highlight that, rather than being an intrinsic limitation of the models, this is a product of the strategy used to compile the prediction set and library (see Methods). We also acknowledge that % MGS predictions for the active molecules were consistently higher than the experimental measurements, albeit accurate within error, for the reasons discussed in Supplementary Section 1.2. Nonetheless, the ensemble approach increases the overall accuracy of the model, and the results presented here for the prediction set clearly justify this approach in retrospect. In regards to the inactive predictions the model also performed very well, with all predictions except two (compounds **11** and **13**) being accurate within the predicted margin of error. The chemical structures of these compounds are shown in Supplementary Fig. S12. Compound **13** was N-methylated isoleucine (Fig. 4b), which was surprisingly active given that N-modifications (methylation and/or acetylation) resulted in a loss of IRI activity for α-alanine and phenylalanine (Supplementary Fig. S13), yet for isoleucine this lowered the % MGS from 19 to 7. Once again, while some of the compounds predicted to be inactive are structurally similar to molecules in the training set, we note that the cyclobutyl moiety found in compounds **10** and **12** is not observed in any compound used during training, highlighting the ability of the model to generalise and predict the IRI activity of new chemical structures with good accuracy.

To rationalise the activity of these molecules in terms of their structures, we computed hydration numbers and indices for all 17 compounds in the prediction set via short MD simulations, as outlined in Methods. Similar to the Amino training set, we observed a positive correlation of up to 0.61 between the hydration indices and % MGS over range of cutoff distances, using either the volume or SASA to normalise by the molecule's size (Supplementary Figs. S17 and S18). We note that this correlation was observed despite not using the hydration indices descriptor to select or rank the prediction set, highlighting the ability of this descriptor to discriminate between active and inactive IRIs effectively.

Given the potent IRI activity of compound **7**, we sought to evaluate its ability to protect RBCs from cold injury during cryopreservation. Despite their critical role in medicine, RBCs are not routinely cryopreserved in clinical settings, as current best practices recommend 20–40 wt% glycerol which must be washed out slowly prior to transfusion causing significant loss of material[24,25]. Using a previously established cryopreservation procedure described in Supplementary Section 2.1 and ref. 26, we found that the addition of 110 mM compound **7** to 15 wt% glycerol offered a significant improvement in post-thaw cell viability of RBCs exposed to transient warming events, providing equivalent protection to 40 wt% glycerol (Supplementary Fig. S21). In contrast, IRI-inactive compound **16** resulted in only limited improvements in cell viability under the same conditions, suggesting a cryoprotective effect resulting from the mitigation of ice growth in vitro.

## Discussion

The discovery of ice recrystallisation inhibitors is of fundamental importance for the development of cryoprotectants that can prevent freezing-induced damage to biological materials. Small molecules offer several advantages, such as biocompatibility, enhanced aqueous solubility and membrane permeability, which render them suitable for off-the-shelf applications in cryopreservation. However, the design of suitable IRI-active molecules has been a daunting task due to a limited knowledge of the molecular mechanisms underlying their activity, and the optimal structural features required for high inhibition.

In this work, we present the first successful application of ML to accurately predict the IRI activity of small molecules. To train ML models, we first generated a dataset by experimental measurements of IRI activity in amino acids, a class of small molecule previously shown to slow ice growth[16,17]. We also curated a dataset of small molecule carbohydrates from literature, with both datasets now made available to facilitate further research in this field. Using these datasets, we applied a data-driven ML approach to predict IRI activity in silico, leveraging different molecular representations that incorporate both fine-grained 3D structural information and solvent interactions derived from MD simulations. This framework was used to perform

virtual screening of a chemical library and select novel IRIs across a spectrum of activity. Having verified these predictions experimentally, we reported the identification of a highly active aminooxazole ester (**7**) that can inhibit ice recrystallisation below 0.5 mg/mL, representing the one of the most IRI-active amino acids discovered to date. Transient warming experiments also revealed a cryoprotective effect of this compound on RBCs, highlighting the utility of this approach to accelerate the discovery of mammalian cell cryoprotectants.

Despite this success, we note that the model's domain of applicability is limited to amino acids or molecules with similar structural features, having found that models trained on the Amino dataset demonstrate poor performance when tasked with predicting the IRI activity of carbohydrates (and vice versa). Combining these datasets during training also resulted in worse performance than training (and testing) on one class of molecule exclusively (Table 3 and Fig. 4). In light of this, we suggest these models be applied to similar materials, with small peptides being one possible direction for the future. Given the small and disparate sources of experimental data in this domain, as well as a number of possible mechanisms underlying IRI, we anticipate that a more generalisable model capable of predicting IRI activity for a wide range of materials is currently far beyond our reach. However, we have provided evidence that models trained using limited data can still achieve excellent performance that can significantly accelerate discovery efforts by moving away from random, trial-and-error selection to rational, ML-empowered decisions.

This work has also demonstrated the predictive power of structural descriptors derived from short MD simulations of small molecules in solution. While more challenging to apply in a prospective virtual screening scenario, such simulations can yield valuable insights into mechanistic behaviours, challenging hypotheses drawn from limited experimental data. Looking ahead, we envisage this ML framework being applied to search libraries of drug-like compounds with known pharmacological and toxicological properties, accelerating their uptake in clinical settings. Altogether, our work highlights the power of data-driven approaches to facilitate the discovery of new cryoprotectants that can address the major challenge of freezing injury during cryopreservation. These molecules are crucial for unlocking the potential of gene- and cell-based therapies, biomanufacturing and cell and tissue banking.

## Methods
### Experimental measurements
Ice recrystallisation (inhibition) was measured using the SCA, as previously described by Knight et al.[27]. A 10 μL drop of each solution was dropped from a height of 1.4 m onto a glass coverslip placed on a thin aluminium plate cooled to −78 °C on dry ice. Upon impact with the coverslip, a polycrystalline ice wafer with an approximate diameter of 10 mm and thickness of 10 μm is formed instantly. The coverslip was then transferred to a Linkam Cryostage BCS196 pre-cooled to −8 °C and left for 30 min at −8 °C to anneal. Photographs were taken after 30 min via a Canon DSLR 500D digital camera using an Olympus CX 41 microscope equipped with a UIS-2 20x/0.45/∞/0-2/FN22 lens and crossed polarisers. The number of crystals in the field of view (FOV) were then counted manually using ImageJ[28], and this number was divided by the FOV area to give the MGS. No automatic image segmentation tools were used, and all images were counted by a single individual to ensure consistency. The MGS for each sample was then compared to a positive control for ice growth, to obtain a % MGS relative to the control. An illustrative example of this procedure is shown in Supplementary Fig. S22. Each experiment was performed in triplicate, and % MGS values were reported as the mean across the three repeats with error bars shown as ± one standard deviation. A further discussion of the measurement confidence and error envelopes is provided in Supplementary Section 1.3.

When performing this assay, it is essential to include saline (or other additives) in the solution to ensure liquid channels form between ice crystals. This allows for ice recrystallisation to occur and prevents false positive results[27]. Typically, using pure water and low concentrations of solutes, no ice growth occurs, generating a false positive. Considering this, a buffer such as phosphate-buffered saline (PBS) is typically employed (with ≈ 140 mM NaCl). However, this concentration of saline is not essential and relevant IRI data has been reported in lower saline concentrations for other materials and small molecules[29,30]. In this work, all compounds in the Amino dataset were assessed for IRI activity at 20 mM using a 10 mM NaCl solution, unless otherwise stated.

### Datasets
In this work, we have used three datasets broadly encompassing two classes of small molecule: amino acids and carbohydrates. The size and distribution of these datasets are shown in Table 1 and Fig. 1. The Glyco and Glyco2 datasets have been compiled using experimental data reported in peer-reviewed literature for a range of small molecule carbohydrates. The Glyco dataset is a subset of Glyco2, and was used in previous work to classify IRI-active small molecules[15]. In addition to the compounds in Glyco, Glyco2 includes an additional 99 structures encompassing greater chemical diversity. This data was obtained from data published in refs. 10,15,20,30–35. The Amino dataset has been assembled via experimental measurements, as previously described in the Methods. Although the same experimental assay was used to evaluate the IRI activity of compounds in the Glyco2 and Amino datasets, different solution conditions were used, meaning the values obtained are not always directly comparable. This issue is discussed further in the Supplementary Section 1.1.

To apply these models to identify novel IRI-active compounds, a set of unseen structures was also required. To obtain this set, we first took the 15 most active and 5 least active compounds in the Amino dataset and screened a database of commercially available compounds (MolPort) for structures bearing similarity to the 20 selections. Similarity was determined by a Tanimoto coefficient[36] >0.7. We chose to include both active and inactive structures in our similarity search to ensure greater coverage of chemical space, whilst providing the opportunity to validate the model's ability to accurately predict both high and low % MGS values. We note that while including only compounds with structural similarity to those in the training set does limit the capacity of the model to identify truly novel IRI-active scaffolds, given the limited number of training examples this approach ensures the prediction set is within the model's domain of applicability. Applying this model to a structurally distinct set of amino acids could be an interesting future study. The prediction set was then filtered to remove any compounds appearing in the training set (i.e. hard overlap), as well as compounds containing co-salts or co-additives, or with a calculated logP > 1.8 (i.e. likely insoluble), computed via RDKit[37]. The final prediction set comprised 497 unique structures. Consensus % MGS predictions for each compounds were then obtained as outlined below. Predictions with a high degree of uncertainty (SD > 15 % MGS) were removed, and the ten most active and ten least active compounds remaining were then purchased and tested experimentally. We note that three compounds from the prediction set were insoluble in aqueous solution at millimolar concentration and excluded from further analysis.

### Molecular representations
We have used six different representations in this work, encompassing low and high-dimensional molecular descriptors. Four of these representations have been applied following successes in a range of tasks such as chemical property prediction (e.g. enthalpies[38], potential energies[39], lipophilicity[40]) and molecular generation[41]:

- 'Standard' descriptors—A collection of ~45 molecular properties that are accessible via RDKit[37], e.g. molecular weight, atom/bond counts, cLogP, topological polar surface area. A complete list of properties is available at https://github.com/gcsosso/DOLMEN.
- Molecular cliques—Considering a molecule as a collection of nodes (atoms) and edges (bonds), i.e. a 2D graph, a clique represents a subgraph of a molecule. A vocabulary of cliques is constructed for a given dataset, and the cliques for each molecule are encoded as a fingerprint. For more information, see ref. 40.
- Histograms of weighted atom-centred symmetry functions (H-wACSF)—Symmetry functions describe the local (3D) chemical environment of an atom in a molecule using radial and angular symmetry functions based on the distance and angles between pairs and triplets of atoms, respectively. In this formulation, element-dependent weighted symmetry functions are computed and the values then binned to obtain a histogram-like descriptor with the same (reduced) dimensionality for all molecules, independent of their size and atomic composition. H-wACSF parameters used here are listed in Supplementary Section 2.2.4; for more detail, see ref. 40.
- Smooth overlap of atomic positions (SOAP) descriptor—3D atomic environments encoded using atomic density fields composed of Gaussian functions centred on each atom. This formalism is extended to describe molecules by averaging the density field across the constituent atoms. SOAP parameters were optimised using a genetic algorithm, as described in ref. 42. The optimal parameter set is provided in Supplementary Section 2.2.3, and readers are directed to refs. 39,42 for more information.

We have also engineered two new and relatively simple molecular descriptors bespoke for this application, via MD simulations:

- Hydration histograms—From short MD simulations, we compute a probability density histogram by calculating the pairwise distances $d$ between each water molecule (specifically the oxygen atom) and all the solute atoms at several points along the trajectory (Supplementary Fig. S14). These distances $d$ are then binned and normalised based on the total number of distances considered and the interval width $\Delta d$, resulting in the probability density $P(d)$ for each bin. Thus, the probability density $P(d)$ for a given distance $d$ is given by $P(d) = \frac{n_d}{\Delta d \sum_i^{D_{cut}} n_i}$, where $n_d$ is the number of water-solute distances in the bin corresponding to the distance $d$, $n_i$ is the number of distances in each bin and $D_{cut}$ is the cutoff distance. Histograms were computed here using 100 bins, up to $d_{cut} = 0.5$ nm, hence $\Delta d = 0.005$ nm.
- Hydration indices—First defined by Tam and colleagues[20], a hydration index represents the number of nonexchangeble water molecules associated with a solute's 'hydration layer' (i.e. hydration number), divided by its partial molar volume. Here, we compute the hydration index computationally by means of MD simulations. Our hydration numbers represent the numbers of waters hydrogen-bonded to the solute, determined via geometric criteria, as well as the numbers of water molecules within a given cutoff distance of the molecule. A hydration index is then obtained by dividing the hydration number by the compound's molecular weight, solvent-accessible surface area (SASA) or molecular volume, all computed via RDKit[37]. The hydration indices descriptor includes ten indices, where the hydration numbers were computed using seven fixed cutoff distances in the range 0.20–0.50 nm, the distances corresponding to the first and second solvation shells (Supplementary Fig. S14) and hydrogen bond numbers.

The standard descriptors and cliques were generated directly from SMILES, while the remaining representations require 3D atomic coordinates for there construction. To generate corresponding 3D conformations, short MD simulations of each system were performed as follows. Each compound was first solvated in water (TIP4P/Ice[43]) in a 4 nm cubic cell and simulated for 20 ns at 273 K via GROMACS 5.1.3[44], using the CHARMM36 forcefield[45]. The final conformation was used to construct H-wACSFs and SOAPs, whereas the hydration descriptors were averaged over 100 different conformations sampled from the trajectories.

Given that many amino acids can exist in multiple ionisation states, it was important to model the relevant form. To determine the predominant state of amino acids under neutral solution conditions, putative ionisation states between the range of pH 6.5 and 7.5 were predicted using the Diamorphite-DL package[46] with a precision factor of 1.0. For structures with multiple predicted states, the final state was selected manually based on $pK_a$ values for the compound found in literature or estimated via the MolGpKa tool[47]. The ionisation states of a random sample were then checked independently to verify the results of this procedure.

## Classification models

For classification, % MGS values were encoded categorically using the activity threshold indicated. All models were trained and evaluated using Keras (Tensorflow)[48], alongside scikit-learn[49]. To perform classification, numerical % MGS values were first converted via one-hot encoding using a defined threshold for activity. The descriptor features were also scaled between 0 and 1, using Min-Max scaling. Independent models were then trained using each of the six representations. A randomised grid search was performed in combination with manual tuning to identify the optimal hyperparameters for each model; these hyperparameters are reported in Supplementary Table 3. Given the small size of the datasets, a leave-one-out (LOO) cross validation (CV) procedure was to train and evaluate each model, in order to obtain robust predictions for each molecule which are independent of training and test splits. 10% of the training data was randomly selected and used as a validation set in conjunction with an early stopping criterion to prevent overfitting. Models were trained over a maximum of 300 epochs, using the binary cross entropy loss function. A classification threshold (probability) of 0.5 was used throughout. In cases where classes were imbalanced, the Synthetic Minority Over-sampling TEchnique (SMOTE) was used to bootstrap the training set to achieve equal number of active/inactive observations. For ensemble classification models, consensus predictions were obtained via a majority voting scheme. In this scheme, the class with the highest number of votes is used. In cases where each class had an equal number of votes, predictions labels defaulted to inactive. All combinations of three or more different descriptor models were explored and ranked based on their F-scores.

## Regression models

Target (% MGS) values and features were first scaled between 0 and 1 using Min-Max scaling. Model hyperparameters (see Supplementary Tables 4 and 5) were identified using a randomised search grid followed by manual tuning. The same LOO CV procedure described above for classification was also used to train and evaluate the regression models, unless otherwise stated. Models were trained over a maximum of 300 epochs, employing the MSE as the loss function. L2 (ridge) regularisation with $\sigma = 0.005$ was also used to prevent overfitting. Ensemble predictions were calculated as the mean predicted value across a given set models, while the standard deviation was used to estimate the associated prediction uncertainty. All combinations of three or more different models were explored and ranked based on prediction accuracy, defined via PCC values.

**Reporting summary**

Further information on research design is available in the Nature Portfolio Reporting Summary linked to this article.

## Data availability

Both the experimental and the computational data that support the findings of this study are available in the Source Data file and the University of Warwick open access research repository (WRAP) with the identifier https://wrap.warwick.ac.uk/187155/[50]. Source data are provided with this paper.

## Code availability

The collection of custom scripts/codes that we utilised in this study are freely available at https://github.com/gcsosso/DOLMEN (permanent https://doi.org/10.5281/zenodo.13150337[51]) and https://github.com/gcsosso/HIN (permanent https://doi.org/10.5281/zenodo.13150400[52]).

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

## Acknowledgements

M.T.W. thanks the MRC for a studentship through the MRC Doctoral Training Partnership in Interdisciplinary Biomedical Research (grant no. MR/S502534/1). M.I.G. thanks the ERC for a Consolidator Grant (866056) and the Royal Society for an Industry Fellowship (191037) joint with Cytivia. We gratefully acknowledge the use of the ARCHER UK National Supercomputing Service (http://www.archer.ac.uk), which we have accessed via the HecBioSim consortium, funded by the EPSRC (grant no. EP/R029407/1). We also gratefully acknowledge the use of Athena at HPC Midlands+, which was funded by the UK Engineering and Physical Sciences Research Council (grant no. EP/P020232/1), via the HPC Midlands+ consortium. We would also like to acknowledge the high-performance computing facilities provided by the Scientific Computing Research Technology Platform at the University of Warwick.

## Author contributions

M.T.W. performed the computational work and analysis. M.T.W., C.I.B., and A.B. performed the experimental work and analysis. All authors interpreted the results. M.T.W., G.C.S. and M.I.G. conceived the research. M.T.W., G.C.S., and M.I.G. wrote the manuscript.

## Competing interests

The authors declare no competing interests.
