## [Peer Review File · Nature Communications]

Data-driven Discovery of Potent Small Molecule Ice Recrystallisation InhibitorsREVIEWER COMMENTS

Reviewer #1 (Remarks to the Author):

In this paper, a machine learning-based IRI prediction model for small molecules is constructed. However, the model has some limitations in terms of interpretability and innovativeness. Firstly, the machine learning prediction model has a high dependence on the quality and quantity of the predicted data. The low explanatory nature of the prediction model used in this paper means that it is difficult to understand how the model makes its predictions. This may have an impact on the stability and accuracy of the model in real-world applications. Secondly, the insufficient amount of data in the dataset as well as the lack of distinctive features may lead to poor performance of the model when generalised to new data. In order to improve the predictive performance of the model, the dataset needs to be further expanded and the saliency of the features needs to be improved through feature engineering and data preprocessing methods. In addition, the machine learning models used in this work were all existing methods without innovations for IRI prediction. In future studies, attempts can be made to introduce more targeted features and novel algorithms to improve the predictive accuracy and interpretability of the models. In conclusion, although this paper has achieved some results in small molecule IRI prediction models, it still needs to be improved in terms of interpretability and innovation. In order to make the model more valuable in practical applications, we need to continuously improve the predictive performance, interpretability and generalisation ability of the model.

Here, a number of small molecules with IRI activity were successfully screened by using machine learning prediction models, and wet experiments were performed to characterise the screened small molecules. However, there are some shortcomings in this study. Firstly, after screening the small molecules with IRI activity, the newly screened small molecules were not compared in terms of their ability with the existing reported small molecules with IRI activity. Such a comparison would help to better assess the activities of the newly screened small molecules as well as their potential value. Second, the mechanism of action of small molecules with IRI activity was not explored in depth in this paper. To remedy these shortcomings, it is suggested that an in-depth study of the changes of small molecules in an ice-water environment be carried out in conjunction with molecular dynamics simulation. Molecular dynamics simulations can provide a detailed understanding of the conformational

changes of small molecules in the ice-water environment and how these changes affect their IRI activity.

Reviewer #2 (Remarks to the Author):

Review of “Data-driven discovery of potent small molecule ice recrystallisation inhibitors,” by M. T. Warren et al. 2024

It was a pleasure to read the preprint “Data-driven discovery of potent small molecule ice recrystallisation inhibitors,” which is a well written and edited account of both computational and experimental investigations of ice recrystallization inhibitors (IRIs). The study’s aim is to use computational, machine learning (ML), tools to identify IRI active molecules using compound libraries, including two publicly available libraries and one library which was generated through freezing experiments conducted for this study. Finally the computational model is tested on a further library that consists of \approx 500 amino acids. The results show that the ML technique is good at predicting IRI compounds, and is successful in identifying compounds that had not be previously labeled as IRI.

While the manuscript is very readable and overall well edited, I believe there are several areas for improvement, especially for the purposes of attracting a broad audience to the manuscript. In its submitted form the manuscript focuses heavily on the machine learning aspects of the author’s studies. My assumption is that this choice of focus was made because this is perhaps the most “novel” contribution of the study, and also perhaps the current popularity of these computational techniques. In this regard I feel the author’s sell themselves short. It appears that the *Amino* dataset, consisting of 63 compounds (Table 1), is the result of experiments, newly conducted for this work. The fact that new experimental results are reported is not mentioned in the abstract, nor in any introduction/motivation text. Moreover, the screening of compounds from an otherwise not utilized compound library is also not really mentioned until deep into the text. Finally, although my ability to evaluate the text from a contribution to Machine Learning point of view is limited (and I would encourage the editors to identify a qualified reviewer in this regard), I find that there is too little attention paid to the potential larger implications/applications of the work. Could the very same techniques identify other important molecular functionalities? What have the author’s learned to inform other such studies? Are their limitations in terms of training data (amount, scope, etc.) that will inform future investigations, or investigators? In its current state this is distilled into one sentence at the end of the discussion.

To be accepted for publication I believe the Abstract most urgently needs to be re-written to encompass both the full scope of what is presented, but also to motivate the larger potential context the work can fit into. Moreover, I think some re-writing of both the introductory and discussion text would highly benefit the manuscript. Perhaps only to reiterate, I found myself having to continuously reframe what I understood the authors had done versus what they were utilizing from sources etc. A better road map would have been useful.

Finally as I note below in my more specific comments, there must be a serious treatment and/or justification of the author’s choice of error envelopes. The single standard deviation shown in key plots already has wide envelopes that would grow to engulf the full data spread if the more typical 95% certainty envelope were chosen. The author’s should address this choice and in particular why they feel their interpretation is robust.

I think if the authors can do these things this paper could be interesting to a wide variety of readers, including those interested in ice-biology interactions, recrystallion processes, computational techniques for identifying biological functionality, and other areas. I would encourage the editor to suggest a revision and resubmission for further evaluation.

Itemized Scientific and Editorial Comments:

Specific Suggestions by Page and/or Section (§):

- (Abstract) see above, but also: “Current programmes....” I do not know if such “programmes” exist, it seems that methods is more the likely meaning here. I find the tense is shifting between present and past here and at times throughout the text, generally one should be careful, as too much shifting makes the text less readable. The use of “leveraged” although trendy I think is not correct (this is an adjective, as in leveraged buyout, not the past tense of leverage).
- (Introduction) The use of “biologics” here and throughout the text is a strange sounding shortening. Biological materials? It seems biologics is used in medicine, so can be argued for, but I would encourage the authors to consider its use.

- (page 3) “these protocols” No protocols are ever specified, so it is a strange choice of phrase.
- (Table 1) I suggest to specify in the table where the libraries come from, an extra column perhaps. This would allow the authors to highlight that Amino is self-generated and first reported here, as I understand. Also, why not add the 500 member amino acid library that is later used to run the blind tests? It could be presented in a way to easily illustrate it is used in a blind test and not as part of training the model.
- (page 9) \sim 500 vs. \approx 500, the first “similar to” is generally used to mean of the same order of magnitude. The later \approx to denote approximately. I guess the later is meant.
- (page 9) a MD not “an MD”
- (Fig. 2) Many of the error bars reported in this figure are quite long, although they only report ± 1 standard deviation. If on the otherhand the 95% uncertainty was reported many would span nearly the entire MGS space. I don’t really see a clear discussion of this in the text. What is the justification for the method of reporting and why should readers see difference as important at this level? This is actually a key point to the entire study, so I think the author’s really need to spend some time convincing readers that differences are meaningful.
- (Discussion) see above.
- (Methods) it seems assumed that the entire FOV is covered by ice? I think it is good to specify this, otherwise one would imagine the crystal number would be normalized by ice area, for example.
- (4.3 Molecular representations) misspelling in section title
- (Supplement, §1.2.3 Experimental comparison) “We therefore found...” This is an extremely weak start to this section I would suggest rephrasing, as written it did not follow from what lay before.
- (Supplement, Fig.2) The color scheme is not intuitive with colder colors representing more strongly positive values, suggest a different color map.
- (Supplement, §2.2.1) Have I missed where TP, FN etc. are defined?
- (Supplement, §2.2.4) 16 and 32 were used
- (Supplement, Fig.17) What information is the reader to take away from the micrograph images? The FOV here is much larger than the ice, so I assume not the same as the above?

Summation: Overall the submitted preprint is quite well written .

Reviewer #3 (Remarks to the Author):

The paper presents a machine learning (ML) model for predicting the Ice Recrystallization Inhibition (IRI) activity of various small molecules. The authors utilized established descriptors and introduced two novel descriptors related to the hydration of the molecules. This work is of significant importance and could pave the way for broader applications of ML methods in the field of cryopreservation. Additionally, the paper demonstrates the application of the ML algorithm to predict the IRI activity of 500 molecules, identifying novel molecules with high IRI activity.

Some observations on the manuscript:

The figures appear too dense and require more space to enhance readability.

Table 3 contains an index "b" under hydration indices, while a footnote references index "a". This inconsistency should be addressed.

While the results are discussed in the results section, I recommend including a discussion of the findings and the limitations of the application in the discussion section. The current discussion resembles more of a conclusion.

It would be beneficial to include additional micrographs from the SCA and provide a brief explanation of the counting methods employed. Clarifying whether any automatic segmentations were used or if all counting was performed manually would improve understanding. (See reference for potential guidance:

<https://doi.org/10.1016/j.cryobiol.2023.02.002> [Titel anhand dieser DOI in Citavi-Projekt übernehmen]).

Overall, I found reading this manuscript enjoyable, and I want to underscore the significance of the approach proposed by the authors.

– Response to Reviewer #1 –

In this paper, a machine learning based IRI prediction model for small molecules is constructed. However, the model has some limitations in terms of interpretability and innovativeness. Firstly, the machine learning prediction model has a high dependence on the quality and quantity of the predicted data. The low explanatory nature of the prediction model used in this paper means that it is difficult to understand how the model makes its predictions. This may have an impact on the stability and accuracy of the model in real-world applications. Secondly, the insufficient amount of data in the dataset as well as the lack of distinctive features may lead to poor performance of the model when generalised to new data. To improve the predictive performance of the model, the dataset needs to be further expanded and the saliency of the features needs to be improved through feature engineering and data pre-processing methods. In addition, the machine learning models used in this work were all existing methods without innovations for IRI prediction. In future studies, attempts can be made to introduce more targeted features and novel algorithms to improve the predictive accuracy and interpretability of the models. In conclusion, although this paper has achieved some results in small molecule IRI prediction models, it still needs to be improved in terms of interpretability and innovation. To make the model more valuable in practical applications, we need to continuously improve the predictive performance, interpretability, and generalisation ability of the model.

Here, a number of small molecules with IRI activity were successfully screened by using machine learning prediction models, and wet experiments were performed to characterise the screened small molecules. However, there are some shortcomings in this study. Firstly, after screening the small molecules with IRI activity, the newly screened small molecules were not compared in terms of their ability with the existing reported small molecules with IRI activity. Such a comparison would help to better assess the activities of the newly screened small molecules as well as their potential value. Second, the mechanism of action of small molecules with IRI activity was not explored in depth in this paper. To remedy these shortcomings, it is suggested that an in-depth study of the changes of small molecules in an ice-water environment be carried out in conjunction with molecular dynamics simulation. Molecular dynamics simulations can provide a detailed understanding of the conformational changes of small molecules in the ice-water environment and how these changes affect their IRI activity.

Our reply: We thank the Reviewer for their comments. The capabilities of our model, as those of any other data-driven model, are indeed limited by both the amount and the quality of the data at our disposal. This is why we have expanded on the data previously available within the recent literature (chiefly [Sci. Rep. 6(1), 26403 (2016)]) by performing ~ 80 sets of new experimental measurements, which yielded 80 additional data points. This represents a significant effort that substantially improves on the limited amount of IRI data available to the community.

In addition, for each of these measurements we have assessed their reliability (see e.g., the inset of Fig. 1a), which in turn allowed us to put into context the results of our predictions – specifically the confidence intervals illustrated in Fig.2. Also, note that we have leveraged ensemble learning techniques to ensure the robustness of our predictions. Indeed, the fact that our model managed to identify several compounds with exceptional IRI activity despite the limited amount of data points at our disposal is a surprising result, that we hope will foster the further expansion of our dataset via future experimental efforts. We have expanded the abstract, the introduction and the conclusions to highlight these aspects in greater detail.

We do agree, to an extent, with the Reviewer re: the lack of interpretability of our model, in that the diverse combination of descriptors we have utilized prevents us from extracting clear guidelines in terms of the rational design of novel IRI from our models as a whole. However, we note that some specific descriptors did give us valuable insight into the structure-function relationship of IRI molecules: the hydration numbers and indices that we have computed are an excellent example in this regard – and in fact we have decided to move the corresponding discussion of these features from the SI into the main text. Re: the two specific issues raised by the Reviewer:

1. “the newly screened small molecules were not compared in terms of their ability with the existing reported small molecules with IRI activity.” We reported the predicted %MGS values (and corresponding uncertainties) for the ~500 newly screened small molecules in Fig. S9 and have now added an additional figure (Fig. S19) to

show the distributions of experimental %MGS values for the prediction set, compared to the training data. In our discussions on page 14, we have also gone into greater detail to put the IRI activity of the newly discovered molecules in the context of previous work. We note that the IRI activity of the most potent inhibitors identified in this work is reported in great detail (e.g., for different concentration of the active molecules in different buffers) in the SI, specifically Fig. S15 and S16.

2. “the mechanism of action of small molecules with IRI activity was not explored in depth in this paper”. The Reviewer is correct – this paper does not seek to provide an in-depth explanation for the mechanism of action of amino acids as IRI agents. That is an entirely different (and rather complex) research question, that we have started to tackle only recently, see e.g., our two recent papers (a): “Minimalistic ice recrystallisation inhibitors based on phenylalanine” [Chem. Comm. 58(55), 7658-7661 (2022)] and, (b): “Ice Recrystallization Inhibition by Amino Acids: The Curious Case of Alpha-and Beta-Alanine” [J. Phys. Chem. Lett. 13, 2237-2244 (2022)]. Note that investigating the mechanism of action of just *two* amino acids (paper (b)), specifically alpha- and beta- alanine, via MD simulations required a substantial computational effort in conjunction with a diverse set of bespoke analysis tools.

Thus, it is unrealistic, at this stage, to embark on a high-throughput investigation of ~80 different amino acids via MD simulations. However, we did perform a new set of MD simulations aimed at investigating the hydration numbers/indices of the *Amino* prediction set. The results are summarised in the figure below (and added to the SI as Fig. S17) and strengthen even further the trends we have previously discussed in the manuscript re: the correlation between hydration indices/numbers and IRI activity.

Fig. S17 Heatmap showing the correlation (PCC) between % MGS values and hydration numbers or hydration indices calculated using different cutoffs and size metrics for the *Amino* prediction set. Hydration indices were obtained by dividing the hydration numbers by either the molecular volume or SASA, as indicated.

We also note that, to demonstrate the applicability of our model and to gain further insight into the mechanism of action of the newly identified IRI agents, we have added in the revised version of the manuscript a set of new results re: the cryopreservation of red blood cells (RBCs). Specifically, we observe marked differences when freezing (and crucially thawing) RBCs in the presence of either compound **7** (predicted and validated to be IRI-active by our model) or compound **16** (predicted and validated to be IRI-inactive by our model). The results in terms of the post-thaw cell viability of RBCs exposed to transient warming events in the presence of the two compounds are summarized in Fig. S21 (also reported below). Strikingly, we find that the addition of 110 mM of compound **7** to 15 % wt glycerol is equivalent to using 40 % wt glycerol (the current tool of the trade in terms of cryopreservation protocols). These results are suggestive of a cryoprotective effect resulting from the mitigation of ice growth *in vitro*.

Fig. S21 Percentage of intact RBCs after transient warming injury utilising different cryosolutions: 40% glycerol, 15% glycerol, and 15% glycerol with either 110 mM, 55 mM, or 6.25 mM of ethyl 2-amino-5-methyl-1,3-oxazole-4-carboxylate (compound **7**, top) and 4,4-dimethylpyrrolidine-2-carboxylic acid (compound **16**, bottom) respectively.

– Response to Reviewer #2 –

It was a pleasure to read the preprint “Data-driven discovery of potent small molecule ice recrystallisation inhibitors,” which is a well written and edited account of both computational and experimental investigations of ice recrystallization inhibitors (IRIs). The study’s aim is to use computational, machine learning (ML), tools to identify IRI active molecules using compound libraries, including two publicly available libraries and one library which was generated through freezing experiments conducted for this study. Finally, the computational model is tested on a further library that consists of \approx 500 amino acids. The results show that the ML technique is good at predicting IRI compounds and is successful in identifying compounds that had not be previously labeled as IRI.

Our reply: We thank the Reviewer for their very positive assessment of our work.

While the manuscript is very readable and overall, well edited, I believe there are several areas for improvement, especially for the purposes of attracting a broad audience to the manuscript. In its submitted form, the manuscript focuses heavily on the machine learning aspects of the author’s studies. My assumption is that this choice of focus was made because this is perhaps the most “novel” contribution of the study, and perhaps the current popularity of these computational techniques. In this regard I feel the author’s sell themselves short. It appears that the Amino dataset, consisting of 63 compounds (Table 1), is the result of experiments, newly conducted for this work. The fact that new experimental results are reported is not mentioned in the abstract, nor in any introduction/motivation text. Moreover, the screening of compounds from an otherwise not utilized compound library is also not really mentioned until deep into the text. Finally, although my ability to evaluate the text from a contribution to Machine Learning point of view is limited (and I would encourage the editors to identify a qualified reviewer in this regard), I find that there is too little attention paid to the potential larger implications/applications of the work. Could the very same techniques identify other important molecular functionalities? What have the author’s learned to inform other such studies? Are their limitations in terms of training data (amount, scope, etc.) that will inform future investigations, or investigators? In its current state this is distilled into one sentence at the end of the discussion. To be accepted for publication I believe the Abstract most urgently needs to be re-written to encompass both the full scope of what is presented, but also to motivate the larger potential context the work can fit into. Moreover, I think some re-writing of both the introductory and discussion text would highly benefit the manuscript. Perhaps only to reiterate, I found myself having to continuously reframe what I understood the authors had done versus what they were utilizing from sources etc. A better road map would have been useful. [] I think if the authors can do these things this paper could be interesting to a wide variety of readers, including those interested in ice-biology interactions, recrystallisation processes, computational techniques for identifying biological functionality, and other areas. I would encourage the editor to suggest a revision and resubmission for further evaluation.*

Our reply: We thank the Reviewer for these useful suggestions. We have now heavily modified the abstract, introduction and discussion to both offer a more logical structure to the reader and to highlight the broad scope of this work. We have clarified that we have indeed performed \sim 80 sets of new experimental measurements, which yielded 80 additional data points. We have also clarified that the screening procedure involved a library of previously unexplored (from the standpoint of IRI activity) compounds. Finally, we have substantially expanded the discussion to put our results into a broader context, including concrete examples of future research avenues such as expanding the current model to incorporate small peptides as potential IRI agents. All these changes (which amount to almost three pages of additional content re: the original version of the manuscript) are highlighted in red in the marked version of the revised manuscript. We also note that we have added new results concerning the survival rate of red blood cells when exposed to some of the compounds we have identified in this work (see the discussion in the main text as well as Fig. S21). This analysis significantly expands, we feel, the scope of our work and strengthen the practical relevance of our model.

[] Finally, as I note below in my more specific comments, there must be a serious treatment and/or justification of the author’s choice of error envelopes. The single standard deviation shown in key plots already*

has wide envelopes that would grow to engulf the full data spread if the more typical 95% certainty envelope were chosen. The authors should address this choice and in particular why they feel their interpretation is robust.

[and]

• (Fig. 2) Many of the error bars reported in this figure are quite long, although they only report ± 1 standard deviation. If on the other hand the 95% uncertainty was reported many would span nearly the entire MGS space. I don't really see a clear discussion of this in the text. What is the justification for the method of reporting and why should readers see difference as important at this level? This is actually a key point to the entire study, so I think the author's really need to spend some time convincing readers that differences are meaningful.

Our reply: We thank the Reviewer for highlighting this important aspect. We have now added a dedicated discussion of the error and confidence intervals relative to the measurements obtained via the “splat cooling” assay in Supplementary Section 1.3.

In short, we acknowledge the measurements obtained using this assay have inherent noise and variability arising from several sources, including the initial number/crystal size which results from a stochastic nucleation process. We also note that larger errors are typically observed for weaker materials, as exemplified in Figure 2a. The choice of this technique was motivated by experimental throughput, as well as a desire to compare against other materials reported in literature, where this technique is widely used.

Throughout this work we performed three experimental measurements for each amino acid, reporting the final quantity as the mean of these three repeats, alongside the standard deviation. This choice is made based on a trade-off between measurement confidence/accuracy and time. To provide support for this decision and uncertainty quantification, we have also performed a retrospective analysis of the experimental measurements of IRI-activity of L-alpha-alanine (the subject of our previous work [J. Phys. Chem. Lett. 13, 2237-2244 (2022)]) – see Fig. S2 in the revised SI, also reported in the next page. From this, we can show that after three measurements the expected average %MGS begins to converge on the (expected) true value, and that taking the average across three measurements is significantly more robust than a single or two measurements.

While one would ideally like to conduct more experimental repeats, performing five measurements would require almost twice as much time per datapoint, which would have significantly limited experimental throughput and thus the size of our dataset. We also highlight that although not entirely satisfactory, the use of three measurements and reporting of the mean \pm one standard deviation or standard error from three experimental repeats is the standard practice in the field.

Fig. S2 Distributions of average % MGS values from a retrospective analysis of measurements obtained using the “splat cooling” assay for L-alpha-alanine. Given six % MGS measurements, we computed the average % MGS using one to five repeats by taking the mean of all possible combinations of the six measurements. Using all six measurements gave an average % MGS = 54.4. The boxes show the interquartile range and median (orange line) of the data.

• (Abstract) see above, but also: “Current programmes....” I do not know if such “programmes” exist, it seems that methods is more the likely meaning here. I find the tense is shifting between present and past here and at times throughout the text, generally one should be careful, as too much shifting makes the text less readable. The use of “leveraged” although trendy I think is not correct (this is an adjective, as in leveraged buyout, not the past tense of leverage).

Our reply: We have re-written the abstract entirely – in that process, we have taken care of this issue.

• (Introduction) The use of “biologics” here and throughout the text is a strange sounding shortening. Biological materials? It seems biologics is used in medicine, so can be argued for, but I would encourage the authors to consider its use.

Our reply: We have limited the usage of “biologics”, albeit some instances do remain – mostly for the benefit of the people working in the medical community, as the Reviewer rightly pointed out.

• (page 3) “these protocols” No protocols are ever specified, so it is a strange choice of phrase.

Our reply: We appreciate the ambiguity of this phrase given that no specific experimental protocols have been provided and have hence removed this from the revised version.

• (Table 1) I suggest to specify in the table where the libraries come from, an extra column perhaps. This would allow the authors to highlight that Amino is self-generated and first reported here, as I understand. Also, why not add the 500 member amino acid library that is later used to run the blind tests? It could be presented in a way to easily illustrate it is used in a blind test and not as part of training the model.

Our reply: We thank the Reviewer for this suggestion – we have implemented it the revised version of Table 1, which now includes the screening library and columns listing the origin of the dataset and their purpose, i.e., training or (blind) testing.

Data-driven Discovery of Potent Small Molecule Ice Recrystallisation Inhibitors Rebuttal

• (page 9) ~500 vs. ≈500, the first “similar to” is generally used to mean of the same order of magnitude. The later ≈ to denote approximately. I guess the later is meant.

Our reply: The Reviewer is correct – we have taken care of this in the revised version of the manuscript.

• (page 9) a MD not “an MD”

Our reply: Fixed, thank you.

• (Methods) it seems assumed that the entire FOV is covered by ice? I think it is good to specify this, otherwise one would imagine the crystal number would be normalized by ice area, for example.

Our reply: We have now clarified this aspect in the main text and added an illustrative example of the procedure to compute the MGS in Fig. S22.

• (4.3 Molecular representations) misspelling in section title

Our reply: Fixed, thank you.

• (Supplement, §1.2.3 Experimental comparison) “We therefore found...” This is an extremely weak start to this section I would suggest rephrasing, as written it did not follow from what lay before.

Our reply: Fixed, thank you.

• (Supplement, Fig.2) The color scheme is not intuitive with colder colors representing more strongly positive values, suggest a different color map.

Our reply: Agreed – we have now switched from blue to red and vice versa.

• (Supplement, §2.2.1) Have I missed where TP, FN etc. are defined? • (Supplement, §2.2.4) 16 and 32 were used

Our reply: We have expanded on Section 2.2.1 of the SI to clarify this issue.

• (Supplement, Fig.17) What information is the reader to take away from the micrograph images? The FOV here is much larger than the ice, so I assume not the same as the above?

Our reply: We thank the reviewer for pointing this out. We acknowledge that for readers outside of the field it might be unclear how to interpret such images. Thus, we have now described the relevant crystal features and explained what can be inferred from these on page 11 of the revised text.

– Response to Reviewer #3 –

The paper presents a machine learning (ML) model for predicting the Ice Recrystallization Inhibition (IRI) activity of various small molecules. The authors utilized established descriptors and introduced two novel descriptors related to the hydration of the molecules. This work is of significant importance and could pave the way for broader applications of ML methods in the field of cryopreservation. Additionally, the paper demonstrates the application of the ML algorithm to predict the IRI activity of 500 molecules, identifying novel molecules with high IRI activity. Overall, I found reading this manuscript enjoyable, and I want to underscore the significance of the approach proposed by the authors.

Our reply: We thank the Reviewer for their very positive assessment of our work.

• The figures appear too dense and require more space to enhance readability.

Our reply: We have divided the original Fig. 2 (which, we agree, was rather crowded) into two separate figures, now Fig. 2 and Fig. 4, so as to improve on the readability of our manuscript.

Data-driven Discovery of Potent Small Molecule Ice Recrystallisation Inhibitors Rebuttal

• *Table 3 contains an index "b" under hydration indices, while a footnote references index "a". This inconsistency should be addressed.*

Our reply: Fixed, thank you.

• *While the results are discussed in the results section, I recommend including a discussion of the findings and the limitations of the application in the discussion section. The current discussion resembles more of a conclusion.*

Our reply: We have substantially expanded the discussion to put our results into a broader context, including concrete examples of future research avenues such as expanding the current model to incorporate small peptides as potential IRI agents. All these changes are highlighted in red in the marked version of the revised manuscript.

• *It would be beneficial to include additional micrographs from the SCA and provide a brief explanation of the counting methods employed. Clarifying whether any automatic segmentations were used or if all counting was performed manually would improve understanding. (See reference for potential guidance: <https://doi.org/10.1016/j.cryobiol.2023.02.002>).*

Our reply: We thank the Reviewer for this suggestion. We have now added the following sentence in the revised version of the manuscript: "No automatic image segmentation tools were used, and all images were counted by a single individual to ensure consistency.". We have also added a dedicated figure illustrating the details of the methodology we have adopted to compute the MGS in SI – see Fig. S22, specifically.

REVIEWERS' COMMENTS

Reviewer #2 (Remarks to the Author):

I find that the resubmitted version of the manuscript is a significant improvement and the author's have clearly spent time to address my (and other's) comments on the original manuscript. This new version is a well-told story and I believe a valuable contribution to the field. Moreover, the author's have improved the supplemental material to a substantial degree. Combined with the manuscript the scientific contribution is much more transparent. I endorse the updated submission for publication.

My only small clerical comment is that I find the last 2 sentences of Section 4.1 confusing. 10 mM solution is used but compounds assessed at 20 mM. I understand with a re-reading that the 10 mM refers to the NaCl solution and the 20 mM to the amino acids, but the phrasing here might be slightly improved.

– Response to Reviewer #2 –

I find that the resubmitted version of the manuscript is a significant improvement, and the authors have clearly spent time to address my (and other's) comments on the original manuscript. This new version is a well-told story, and I believe a valuable contribution to the field. Moreover, the authors have improved the supplemental material to a substantial degree. Combined with the manuscript the scientific contribution is much more transparent. I endorse the updated submission for publication.

Our reply: We thank the Reviewer.

My only small clerical comment is that I find the last 2 sentences of Section 4.1 confusing. 10 mM solution is used but compounds assessed at 20 mM. I understand with a re-reading that the 10 mM refers to the NaCl solution and the 20 mM to the amino acids, but the phrasing here might be slightly improved.

Our reply: We thank the Reviewer for this comment. The last two sentences of Section 4.1 have now been rephrased for greater clarity.